# CoVR: Learning Composed Video Retrieval from Web Video Captions

## Abstract

Composed Image Retrieval (CoIR) has recently gained popularity as a task that considers *both* text and image queries together, to search for relevant images in a database. Most CoIR approaches require manually annotated datasets, containing image-text-image triplets, where the text describes a modification from the query image to the target image. However, manual curation of CoIR triplets is expensive and prevents scalability. In this work, we instead propose a scalable automatic dataset creation methodology that generates triplets given video-caption *pairs*. To this end, we mine paired videos with a similar caption from a large database, and leverage a large language model to generate the corresponding modification text. We automatically construct our WebVid-CoVR dataset by applying this procedure to the large WebVid2M collection, resulting in 1.6M triplets. Moreover, we introduce a new benchmark for composed *video* retrieval (CoVR) and contribute a manually annotated evaluation set, along with baseline results. We further show that training a CoVR model on our dataset transfers well to CoIR, improving the state of the art in the zero-shot setup on both the CIRR and FashionIQ benchmarks. Our code, datasets, and models will be made publicly available.

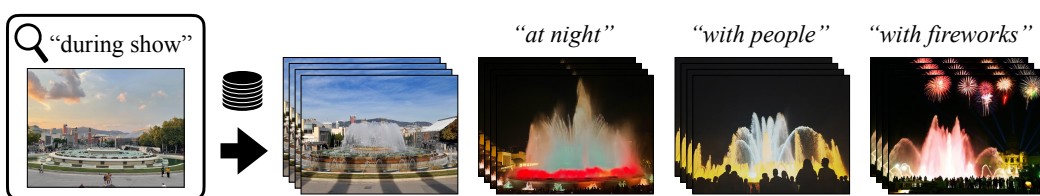

Figure 1: **Task:** Composed Video Retrieval (CoVR) seeks to retrieve *videos* from a database by searching with both a query image and a query text. The text typically specifies the desired modification to the query image. In this example, a traveller might wonder how the photographed place looks like during a fountain show, by describing several modifications, such as "during show at night, with people, with fireworks".

## 1 Introduction

Consider the scenario where a traveller takes a picture of a landmark or scenic spot and wants to discover videos that capture the essence of that location, by specifying certain conditions via text. For example, the query image in Figure 1 (of a fountain in Barcelona), along with the text "during show" should bring the video showcasing the fountain show. Further refining the text query such as "during show at night", would allow the traveller to decide whether to wait for the show until the night time. In this work, our goal is composed video retrieval (CoVR), where the user performs such multi-modal search, by querying an image of a particular visual concept and a modification text, to find videos that exhibit the similar visual characteristics with the desired modification, in a dynamic context.

Submitted to 37th Conference on Neural Information Processing Systems (NeurIPS 2023). Do not distribute.

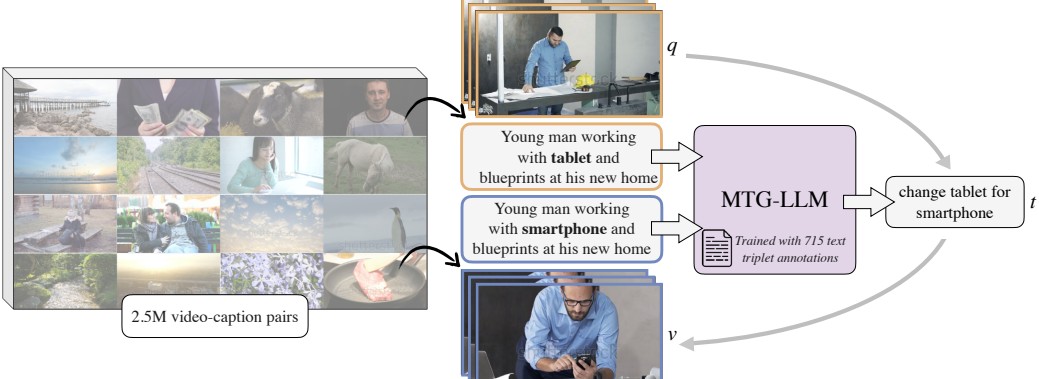

Figure 2: **Method overview:** We automatically mine similar caption pairs from a large video-caption database from the Web, and use our modification text generation language model (MTG-LLM) to describe the difference between the two captions. MTG-LLM is trained on a dataset of 715 triplet text annotations [8]. The resulting triplet of two corresponding videos (query $q$ and target video $v$) and the modification text ($t$) is therefore obtained fully automatically, allowing a scalable CoVR training data generation.

CoVR has many use cases, including but not limited to searching online videos for finding reviews of a specific product, how-to videos of a tool for specific usages, live events in specific locations, sports matches of specific players. Similar to composed image retrieval (CoIR), CoVR is also particularly useful when conveying a concept with a visual is easier and/or more accurate than only using words (e.g., unknown location/object, a specific camera view, a specific color).

Given the increased momentum in vision and language research in the recent years [31, 45], CoIR has emerged as a new task [57], and since then witnessed improvements of both models and benchmarks [6, 7, 21, 28, 37, 58]. However, to the best of our knowledge, CoVR was not studied before. A key challenge in building CoVR models is the difficulty of gathering suitable training data of image-text-video triplets. We overcome this limitation by developing an automatic approach to generate triplets from existing video-caption collections. Specifically, we mine video pairs whose corresponding captions slightly differ in text space. We automatically describe this difference with a language model, which we train for a *modification-text generation* task. In particular, we use manually annotated triplets, each containing: (a) source caption, (b) target caption, (c) the modification text. We then finetune a large language model (LLM) [54] by inputting (a-b), and outputting (c). We assume the resulting modification to describe the difference between the corresponding videos, thus obtaining video-text-video triplets (see Figure 2 for an overview). When training our CoVR/CoIR models, we can select one or more frames from the videos, enabling multiple settings (i.e., retrieving images or videos).

We apply our triplet generation approach to the WebVid2M dataset [4] which contains 2.5M Web-scraped video-caption pairs. This results in the WebVid-CoVR training dataset with 1.6M CoVR triplets. By virtue of its automatic generation procedure, WebVid-CoVR is inherently noisy. To efficiently train on such large-scale and noisy training data, we use a contrastive loss [55] and additionally sample hard negatives that have the same source caption but different target captions. We design a CoVR model based on the cross-modal BLIP [31] and use query scoring [5] to exploit information from multiple video frames. Training this model on WebVid-CoVR transfers well to the CoIR task, in both zero-shot and finetuning settings, and achieves state-of-the-art results on the CIRR and FashionIQ benchmarks in the zero-shot setup. Finally, to foster research in CoVR, we repeat our generation procedure on a separate subset of the WebVid10M dataset [4] and manually select correctly generated samples to constitute WebVid-CoVR$_m$, a test set of 2,435 CoVR triplets. We find that our model achieves promising results on WebVid-CoVR$_m$ compared to standard baselines.

To summarize, our contributions are: (i) We propose a scalable approach to automatically generate composed visual retrieval training data. We apply this pipeline to the WebVid2M dataset and generate the WebVid-CoVR training dataset with 1.6M CoVR triplets. (ii) We show that training a CoVR model on WebVid-CoVR transfers well to the CoIR task, and achieves state-of-the-art results on the CIRR and FashionIQ benchmarks in the zero-shot setup. (iii) We evaluate our model on WebVid-

Table 1: **Existing datasets:** We compare our proposed WebVid-CoVR training dataset and its manually annotated test set WebVid-CoVR$_m$ with existing composed visual retrieval datasets. 📷 denotes image, 🎥 denotes video datasets. We contribute the largest training dataset for the natural domain. Note that, while SynthTriplets18M is larger, the transfer performance to real images is ineffective potentially due to a domain gap (see Table 3).

| Dataset | Type | #Triplets | #Visuals | #Unique words | Avg. text length | Domain |
|---------|------|-----------|----------|---------------|------------------|--------|
| CIRR [37] | 📷 | 36,554 | 21,185 | 7,129 | 59.51 | Natural |
| FashionIQ [58] | 📷 | 30,132 | 7,988 | 4,425 | 27.13 | Fashion |
| CIRCO [6] | 📷 | 1,020 | - | - | - | Natural |
| LaSCo [28] | 📷 | 389,305 | 121,479 | 13,488 | 30.70 | Natural |
| SynthTriplets18M [21] | 📷 | 18,000,000 | - | - | - | Synthetic |
| WebVid-CoVR | 🎥 | 1,648,789 | 130,775 | 19,163 | 23.36 | Natural |
| WebVid-CoVR$_m$ | 🎥 | 2,435 | 2,435 | 1,764 | 22.03 | Natural |

CoVR$_m$, a new CoVR benchmark that we manually annotate. Our code and dataset are provided in the Supplementary Material, and will be publicly released together with our models.

## 2 Related Work

**Composed image retrieval (CoIR).** CoIR [57] has been an active area of research in recent years [7, 14, 25]. Most methods designed for this problem use manually annotated data for training. Some recent works, such as Pic2Word [47] and SEARLE [6], explore zero-shot CoIR setups where no manually annotated CoIR triplet is used. These approaches build on CLIP [45] and train directly on unlabeled image(-text) data. In contrast, we use unlabeled video-text pairs to automatically generate composed video retrieval (CoVR) triplets, train a CoVR model on the generated data, and study zero-shot and finetuning transfer of the resulting model on both CoIR and CoVR.

**Datasets for composed image retrieval.** CIRR [37] and Fashion-IQ [58] are the two most widely used CoIR benchmarks. Both are manually annotated, hence small scale (about 30K triplets, see Table 1) due to the high cost implied in collecting CoIR triplets. To scale up, two concurrent works proposed larger, automatically generated CoIR datasets: LaSCo [28] and SynthTriplets18M [21]. However, these two datasets are currently not publicly available. The LaSCo dataset [28] is generated using the visual question answering annotations and the pairing between images and counterfactual images in the VQAv2 dataset [3]. In detail, this dataset provides for each (image, question, answer) triplet a counterfactual triplet with the same question and different image and answer. In contrast, we do not rely on such expensive annotation schemes. SynthTriplets18M [21] uses the text-conditioned image editing framework InstructPix2Pix [8] to automatically generate CoIR data. Their edit text generation process is similar to ours, but our generation process differs in that we automatically mine similar videos from a dataset of unlabeled video-text pairs to construct CoVR triplets instead of generating visual data. In experiments, we show the superiority of our generation procedure as we achieve much higher CoIR results (e.g., 38% vs 19% zero-shot R@1 on CIRR while generating fewer data). Lastly, our WebVid-CoVR dataset is composed of videos, and not limited to still images.

**Vision-language pretraining.** Many strong multi-modal models have been pretrained on large datasets of image-caption pairs [2, 13, 24, 27, 30, 32, 34, 38, 45, 48, 51, 67, 71] or video-caption pairs [1, 29, 33, 41, 42, 53, 59, 60, 68, 69, 70]. In contrast, we generate CoVR training data from video-caption pairs instead of directly training on them. Our data generation approach is also related to other generation approaches used for other tasks, e.g., action recognition [43], visual question answering [62, 63] and visual dialog [35]. However, unlike all these tasks, the CoVR task requires retrieving visual data.

**Video retrieval.** Text-to-video retrieval has received great attention over the last few years [17, 18, 19, 36, 39, 40, 46, 59, 61, 64, 65]. We also make use of multiple video frames with query scoring similar to [5]. However, different from these methods, we focus on *composed* video retrieval, where the query consists of both text and visual data.

# 3 Automatic Triplet Generation and CoVR Training

The goal of the composed video retrieval (CoVR) task is, given an input video or image $q$ and a modification text $t$, to retrieve a modified video $v$ in a large database of videos. We wish to avoid the manual annotation of $(q, t, v)$ triplets for training. Hence we automatically generate such triplets from Web-scraped video-caption pairs, as explained in Section 3.1 and illustrated in Figure 2. The resulting WebVid-CoVR dataset, together with its manually curated evaluation set, is presented in Section 3.2. Finally, we present how we train a CoVR model using WebVid-CoVR in Section 3.3.

## 3.1 Generating composed video retrieval triplets

Given a large (Web-scraped) dataset of video-caption pairs $(v, c)$, we wish to automatically generate video-text-video CoVR triplets $(q, t, v)$ where the text $t$ describes a modification to the visual query $q$. However, the dataset of video-caption pairs neither contains annotations of paired videos, nor modification text that describes their difference. Hence we propose a methodology to automatically mine paired videos and describe their difference, as described below. Note that for illustration, we take as an example the WebVid2M dataset [4] with 2.5M video-caption pairs, but this methodology could be applied to other large datasets of video-text (or image-text) pairs.

**Mining paired videos by pairing captions.** In order to obtain paired videos, we leverage their captions. The core idea is that videos with similar captions are likely to have similar visual content. Specifically, we consider captions that differ by a single word, excluding punctuation marks. For instance, the caption *"Young woman smiling"* is paired with *"Old woman smiling"* and *"Young couple smiling"*. In the 2M distinct captions from WebVid2M, this process allows us to identify a vast pool of 1.2M distinct caption pairs with 177K distinct captions, resulting in 3.1M paired videos.

**Filtering caption pairs.** We wish to automatically generate the modification text between paired videos using their (paired) captions. However, caption pairs with the same meaning are likely to result in meaningless differences. On the contrary, caption pairs that differ too much are likely to result in large visual differences that cannot be easily described. To address these issues, we filter out caption pairs that are too similar and too dissimilar. Specifically, we exclude caption pairs with CLIP text embedding similarity $\geq 0.96$ (e.g., *"Fit and happy young couple playing in the park"* and *"Fit and happy young couple play in the park"*) and caption pairs with CLIP text embedding similarity $\leq 0.6$ (e.g., *"Zebra on a white background"* and *"Coins on a white background"*). We also exclude pairs where the captions differ by a digit (which mostly consist of date in practice), or by an out-of-vocabulary word. Finally, we remove templated captions such as *"abstract of"*, *"concept of"*, and *"flag of"* which are over-represented.

**Generating a modification text from paired captions.** In order to generate a modification text between paired videos, we apply a modification text generation large language model (MTG-LLM) to their corresponding paired captions. We describe the MTG-LLM inference process below and then explain its training details. The MTG-LLM takes as input two paired captions and generates a modification text that describes the difference between the two captions (see Fig. 2). In detail, the generation is auto-regressive, i.e., we recursively sample from the token likelihood distribution conditioned on the previously generated tokens until an end-of-sentence token is reached. To increase the diversity of the generated samples, we use top-k sampling instead of maximum-likelihood-based methods such as beam search and its variants [56]. Note that we only generate a single modification text per caption pair for computational efficiency, but the MTG-LLM could be used to generate multiple modification texts per caption pair which could serve as a data augmentation in future work.

We now describe the training details of the MTG-LLM. We start from a LLM pretrained with a next token prediction objective on a Web-scale text dataset [54]. We then finetune this LLM for the MTG task on a manually annotated text dataset. In particular, we repurpose the editing dataset from InstructPix2Pix [8], which provides a modification text and a target caption for 700 input captions. We augment this dataset with 15 additional annotations that are useful in our use case. These examples involve transformations such as changing singular nouns to plural (*tree* to *trees*), as well as addressing specific edge cases. More details can be found in the Supplementary Material.

**Filtering video pairs.** We wish to avoid some modification texts being over-represented in the dataset as it could harm training. Hence, if there are more than 10 video pairs associated with the same pair of captions (therefore leading to the same modification text), we only select 10 video pairs. As the CoVR task typically involves similar query-target video pairs, we choose pairs of videos with

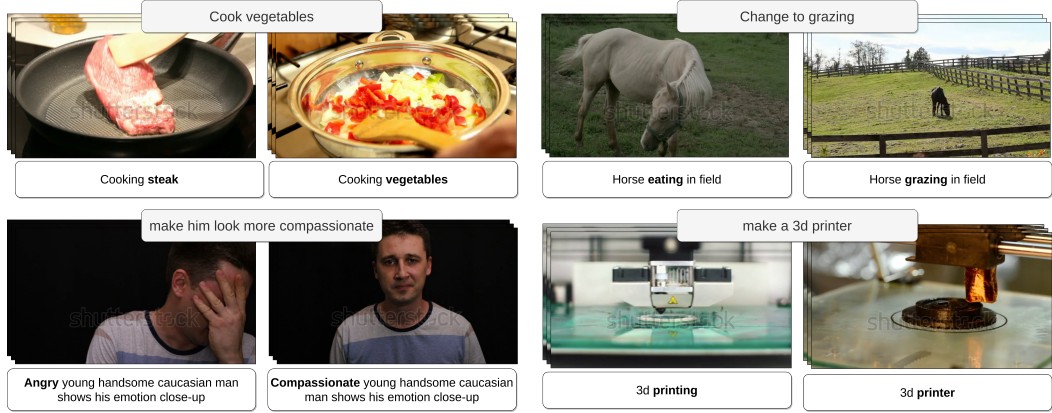

Figure 3: **Examples of generated CoVR triplets in WebVid-CoVR:** The middle frame of each video is shown with its corresponding caption, with the distinct word highlighted in bold. Additionally, the generated modification text is displayed on top of each pair of videos.

the highest visual similarity, as measured by the CLIP visual embedding similarity computed at the middle frame of the videos.

## 3.2 Analysis of WebVid-CoVR

**WebVid-CoVR: a large-scale CoVR training dataset.** By applying the previously described pipeline to the WebVid2M dataset [4], we generate WebVid-CoVR, a dataset containing 1.6M CoVR triplets, which is significantly more than prior datasets (see Table 1). On average, a video lasts 16.8 seconds, a modification text contains 4.8 words, and one target video is associated with 12.7 triplets. WebVid-CoVR is highly diverse with 131K distinct videos and 467K distinct modification texts. Examples of CoVR triplets from the WebVid-CoVR dataset are illustrated in Figure 3. These examples show the diversity of the data in WebVid-CoVR, and its noise due to the automatic generation procedure. We provide further analysis of the WebVid-CoVR dataset in the supplementary material.

**WebVid-CoVR$_m$: a new CoVR evaluation benchmark.** Due to the noise in WebVid-CoVR, we manually annotate a small test set, dubbed WebVid-CoVR$_m$, for evaluation. For this, we first repeat the data generation procedure described in Section 3.1, but on a different corpus of video-caption pairs. Specifically, we consider video-caption pairs from the WebVid10M corpus [4] that are not included in the WebVid2M dataset, resulting in a pool of 8 million video-caption pairs. This ensures that other models using WebVid2M for pretraining have not been exposed to any of the test examples. In the video pairs filtering stage, for each pair of captions, we here only keep one pair of videos (the one with the highest visual similarity). This results in 163K candidate triplets that could be used for testing purposes. We randomly sample 7K triplets that we use for validation and randomly sample 3.1K other triplets that we manually annotate as described below.

We augment the 3.1K triplets by generating two additional modification texts with the MTG-LLM. The annotator reads the three generated modification texts, looks at three frames from the query and target videos, and either keeps the best modification text if at least one is valid or discards the sample. Through this meticulous annotation process, we ensure that the test set comprises high-quality and meaningful CoVR triplets. This results in a test set of 2.4K triplets, i.e., about 23% of the examples are considered as noisy and are discarded.

## 3.3 Training on WebVid-CoVR

Here, we describe our CoVR model architecture and how we train it on our WebVid-CoVR dataset.

**CoVR-BLIP model architecture.** Our model architecture builds upon a pretrained image-text model, BLIP [31]. The BLIP model is pretrained on a large dataset of image-caption pairs with three vision-language objectives: image-text contrastive learning, image-text matching, and image-conditioned language modeling. However, BLIP is not pretrained for composed visual retrieval with both visual and text inputs. Therefore we adapt BLIP to the CoIR/CoVR task as follows.

We use the BLIP image encoder to encode the image query. The resulting visual features and the modification text are then forwarded to the BLIP image-grounded text encoder together, which outputs a multi-modal embedding $f_i \in \mathbb{R}^d$ where $d$ is the embedding dimension. To retrieve a target video from a database of videos $V$, we compute embedding vectors for all possible videos as follows. We uniformly sample $N$ frames from the video and compute a weighted mean of the BLIP image embeddings to obtain the video embedding vector $\hat{v} \in \mathbb{R}^d$. The weights are obtained by computing the image-caption similarity for every video frame with BLIP image and text encoder, respectively, similar to [4] in the context of text-to-video retrieval. Finally, given a multi-modal embedding $f_i$, the retrieved video is the one that maximizes the embedding similarity, i.e., $\arg\max_{v \in V}(\hat{v} . f_i^T)$.

**Training.** In order to train on WebVid-CoVR, we use a contrastive learning approach [44, 55], as it has been shown to be effective to learn strong multi-modal representations from large-scale noisy data [41, 45]. We make several design choices to maximize its efficiency. First, we create a training batch by sampling distinct target videos and for each target video, we randomly sample an associated query image and modification text. This ensures that the same target video appears only once in a batch and maximizes the number of different target videos that can be used as negatives in contrastive learning.

Second, following HN-NCE [44], we use as negatives all target videos $v_{j \in \mathcal{B}}$ in the batch $\mathcal{B}$ and additionally increase the weight of most similar samples. In addition, we mine hard negative samples that we select based on the captions associated with the videos in WebVid2M. Specifically, for a given $(q_i, t_i, v_i)$ triplet, we consider as hard negatives all instances in the batch $(q_j, t_j, v_j) \in HN(i)$ where $q_i$ and $q_j$ have the same caption but $v_i$ and $v_j$ have different captions. In addition, to reduce the number of noisy negatives with the same semantic content as a given sample $i$, we exclude from the computation of the loss samples $(q_j, t_j, v_j) \in P(i)$ for which $v_i$ and $v_j$ have the same caption.

Formally, given a training batch $\mathcal{B}$ of triplets $(q_i, t_i, v_i)$, we minimize the following loss:

$$\mathcal{L}(\mathcal{B}) = \sum_{i \in \mathcal{B}} \{ \; -\log(\frac{e^{S_{i,i}/\tau}}{\sum_{j \in \mathcal{B} \setminus P(i)} e^{S_{i,j}/\tau} w_{i,j} + \alpha \sum_{j \in HN(i)} e^{S_{i,j}/\tau}})$$
$$-\log(\frac{e^{S_{i,i}/\tau}}{\sum_{j \in \mathcal{B} \setminus P(i)} e^{S_{j,i}/\tau} w_{j,i} + \alpha \sum_{j \in HN(i)} e^{S_{j,i}/\tau}}) \}$$

where $\alpha$ and $\tau$ are learnable parameters, $S_{i,j}$ is the cosine similarity between the multi-modal embedding $f_i$ and the target video embedding $\hat{v}_i$, $HN(i)$ is the set of hard negatives, $P(i)$ is the set of noisy negatives and $w_{i,j}$ is set as in [44].

# 4 Experiments

In this Section, we first describe the experimental protocol including the datasets, evaluation metrics, and implementation details (Section 4.1). We then present the results of CoVR on our new video benchmark (Section 4.2), as well as transfer results of CoIR on standard image benchmarks (Section 4.3). Finally, we provide ablations on our key components (Section 4.4).

## 4.1 Experimental setup

**Datasets.** WebVid-CoVR is our proposed training CoVR dataset presented in Section 3.2, and WebVid-CoVR$_m$ is our new CoVR benchmark presented in Section 3.2.

CIRR [37] is a manually annotated CoIR dataset that contains open-domain natural images from NLVR2 [52]. It contains 36.5K queries annotated on 19K different images. CIRR includes two benchmarks: a standard one with the target search space as the entire validation corpus, and a fine-grained *subset*, where the search space is a subgroup of six images similar to the query image (based on pretrained ResNet15 feature distance). The dataset is divided into training, validation, and testing splits with 28,225/16,742, 4,181/2,265 and 4,148/2,178 queries/images, respectively.

FashionIQ [58] is a CoIR dataset that contains images of fashion products, divided into three categories of Shirts, Dresses, and Tops/Tees. The query and target images were automatically paired based on title similarities (crawled from the web), and modification texts were then manually annotated. This dataset consists of 30K queries annotated on 40.5K different images. It is divided into training and validation splits with 18,000/45,429 and 6,016/15,415 queries/images, respectively.

Table 2: **Benchmarking on the WebVid-CoVR$_m$ test set:** We find that training on WebVid-CoVR, using both the visual and text input modalities, and using multiple frames to model the target video are all important factors of CoVR performance.

| Train on WebVid-CoVR | Method | Input modalities | #frames | R@1 | R@5 | R@10 | R@50 |
|---|---|---|---|---|---|---|---|
| No | Random | - | - | 0.08 | 0.21 | 0.49 | 2.34 |
| | CoVR-BLIP | Text | - | 19.88 | 37.66 | 45.91 | 66.08 |
| | CoVR-BLIP | Visual | 15 | 37.04 | 61.36 | 69.94 | 87.23 |
| | CoVR-BLIP | Visual+Text | 15 | 15.98 | 33.22 | 41.36 | 59.18 |
| Yes | CoVR-BLIP | Text | - | 20.78 | 41.68 | 51.29 | 71.05 |
| | CoVR-BLIP | Visual | 15 | 37.04 | 61.36 | 69.94 | 87.23 |
| | CoVR-BLIP | Visual+Text | 1 | 53.43 | 80.00 | 87.27 | 97.66 |
| | CoVR-BLIP | Visual+Text | 15 | **54.87** | **80.99** | **88.30** | **98.11** |

Table 3: **State-of-the-art comparison on the CIRR test set:** Our model benefits from training on WebVid-CoVR in the zero-shot setting, and in the finetuning setting where it performs competitively. † denotes results reported by [37].

| Mode | Method | Pretraining Data | Recall@K | | | | R$_{subset}$@K | | |
|---|---|---|---|---|---|---|---|---|---|
| | | | K=1 | K=5 | K=10 | K=50 | K=1 | K=2 | K=3 |
| Train (CIRR) | TIRG [57]† | - | 14.61 | 48.37 | 64.08 | 90.03 | 22.67 | 44.97 | 65.14 |
| | TIRG+LastConv [57]† | - | 11.04 | 35.68 | 51.27 | 83.29 | 23.82 | 45.65 | 64.55 |
| | MAAF [15]† | - | 10.31 | 33.03 | 48.30 | 80.06 | 21.05 | 41.81 | 61.60 |
| | MAAF-BERT [15]† | - | 10.12 | 33.10 | 48.01 | 80.57 | 22.04 | 42.41 | 62.14 |
| | MAAF-IT [15]† | - | 9.90 | 32.86 | 48.83 | 80.27 | 21.17 | 42.04 | 60.91 |
| | MAAF-RP [15]† | - | 10.22 | 33.32 | 48.68 | 81.84 | 21.41 | 42.17 | 61.60 |
| | ARTEMIS [14] | - | 16.96 | 46.10 | 61.31 | 87.73 | 39.99 | 62.20 | 75.67 |
| | CIRPLANT [37] | - | 19.55 | 52.55 | 68.39 | 92.38 | 39.20 | 63.03 | 79.49 |
| | LF-BLIP [7, 28] | - | 20.89 | 48.07 | 61.16 | 83.71 | 50.22 | 73.16 | 86.82 |
| | CompoDiff [21] | SynthTriplets18M [21] | 22.35 | 54.36 | 73.41 | 91.77 | 35.84 | 56.11 | 76.60 |
| | Combiner [7] | - | 33.59 | 65.35 | 77.35 | 95.21 | 62.39 | 81.81 | 92.02 |
| | CASE [28] | - | 48.00 | 79.11 | 87.25 | **97.57** | 75.88 | **90.58** | **96.00** |
| | CASE [28] | LaSCo [28] | 48.68 | 79.98 | 88.51 | 97.49 | 76.39 | 90.12 | 95.86 |
| | CASE [28] | LaSCo [28]+COCO [10] | 49.35 | **80.02** | **88.75** | 97.47 | **76.48** | 90.37 | 95.71 |
| | CoVR-BLIP | - | 49.33 | 78.51 | 86.53 | 94.53 | 75.81 | 88.29 | 92.99 |
| | CoVR-BLIP | WebVid-CoVR | **50.55** | 79.23 | 87.30 | 94.70 | 75.69 | 88.58 | 93.33 |
| Zero Shot | Random† | - | 0.04 | 0.22 | 0.44 | 2.18 | 16.67 | 33.33 | 50.00 |
| | CompoDiff [21] | SynthTriplets18M [21] | 19.37 | 53.81 | 72.02 | 90.85 | 28.96 | 49.21 | 67.03 |
| | Pic2Word [47] | Conceptual Captions [49] | 23.90 | 51.70 | 65.30 | 87.80 | - | - | - |
| | CASE [28] | LaSCo [28] | 30.89 | 60.75 | 73.88 | 92.84 | 60.17 | 80.17 | 90.41 |
| | CASE [28] | LaSCo [28]+COCO [10] | 35.40 | 65.78 | **78.53** | **94.63** | 64.29 | 82.66 | **91.61** |
| | CoVR-BLIP | - | 19.76 | 41.23 | 50.89 | 71.64 | 63.04 | 81.01 | 89.37 |
| | CoVR-BLIP | WebVid-CoVR | **38.55** | **66.80** | 77.25 | 91.61 | **69.42** | **84.22** | 91.16 |

**Evaluation metrics.** Following standard evaluation protocols [37], we report the video retrieval recall at rank 1, 5, 10, and 50. Recall at rank k (R@k) quantifies the number of times the correct video is among the top k results. MeanR denotes the average of R@1, R@5, R@10, and R@50. Higher recall means better performance.

**Implementation details.** For our MTG-LLM, we use LLaMA 7B model [54] that we finetune for one epoch with an initial learning rate of $3e^{-5}$ for MTG. For our CoVR model, we use the BLIP with ViT-L [16] at 384 pixels finetuned for text-image retrieval on COCO and freeze the ViT for computational efficiency. We train our CoVR model on WebVid-CoVR for 3 epochs with a batch size of 2048 and an initial learning rate of $1e^{-5}$. To finetune on CIRR/FashionIQ, we train for 6/3 epochs with a batch size of 2048/1024 and an initial learning rate of $5e^{-5}/1e^{-4}$. Experiments are conducted on 4 NVIDIA A100-SXM4-80GB GPUs. More details are included in the Supplementary Material.

## 4.2 Composed video retrieval results

We report CoVR results on our WebVid-CoVR$_m$ test set in Table 2. For models trained on WebVid-CoVR, we find that using both modalities is crucial for performance, as the model with visual and text inputs outperforms both the text-only and the visual-only models. Furthermore, using multiple target video frames is beneficial, as the model with 15 frames improves over the model with 1 frame.

Table 4: **State-of-the-art comparison on the FashionIQ validation set:** Our model benefits from training on WebVid-CoVR in the zero-shot setting, and in the finetuning setting. CC3M is Conceptual Captions 3M [9].

| Mode | Method | Pretraining Data | Shirt R@10 | Shirt R@50 | Dress R@10 | Dress R@50 | Toptee R@10 | Toptee R@50 | Average R@10 | Average R@50 |
|---|---|---|---|---|---|---|---|---|---|---|
| | JVSM [11] | - | 12.0 | 27.1 | 10.7 | 25.9 | 13.0 | 26.9 | 11.9 | 26.6 |
| | CIRPLANT [37] | - | 17.53 | 38.81 | 17.45 | 40.41 | 61.64 | 45.38 | 18.87 | 41.53 |
| | TRACE w/BER [23] | - | 20.80 | 40.80 | 22.70 | 44.91 | 24.22 | 49.80 | 22.57 | 46.19 |
| | VAL w/GloVe [12] | - | 22.38 | 44.15 | 22.53 | 44.00 | 27.53 | 51.68 | 24.15 | 46.61 |
| | MAAF [15] | - | 21.3 | 44.2 | 23.8 | 48.6 | 27.9 | 53.6 | 24.3 | 48.8 |
| | CurlingNet [66] | - | 21.45 | 44.56 | 26.15 | 53.24 | 30.12 | 55.23 | 25.90 | 51.01 |
| | RTIC-GCN [50] | - | 23.79 | 47.25 | 29.15 | 54.04 | 31.61 | 57.98 | 28.18 | 53.09 |
| | CoSMo[26] | - | 24.90 | 49.18 | 25.64 | 50.30 | 29.21 | 57.46 | 26.58 | 52.31 |
| Train | ARTEMIS[14] | - | 21.78 | 43.64 | 27.16 | 52.40 | 29.20 | 53.83 | 26.05 | 50.29 |
| (FashionIQ) | DCNet[25] | - | 23.95 | 47.30 | 28.95 | 56.07 | 30.44 | 58.29 | 27.78 | 53.89 |
| | SAC w/BERT[22] | - | 28.02 | 51.86 | 26.52 | 51.01 | 32.70 | 61.23 | 29.08 | 54.70 |
| | FashionVLP[20] | - | 31.89 | 58.44 | 32.42 | 60.29 | 38.51 | 68.79 | 34.27 | 62.51 |
| | LF-CLIP (Combiner) [7] | - | 36.36 | 58.00 | 31.63 | 56.67 | 38.19 | 62.42 | 35.39 | 59.03 |
| | LF-BLIP [7, 28] | - | 25.39 | 43.57 | 25.31 | 44.05 | 26.54 | 44.48 | 25.75 | 43.98 |
| | CASE [28] | LaSCo [28] | **48.48** | **70.23** | **47.44** | **69.36** | 50.18 | 72.24 | 48.79 | **70.68** |
| | CoVR-BLIP | - | 48.04 | 68.20 | 44.92 | 68.91 | 52.47 | **74.71** | 48.48 | 70.61 |
| | CoVR-BLIP | WebVid-CoVR | **48.48** | 67.86 | 45.31 | 68.37 | **53.14** | 73.94 | **48.98** | 70.06 |
| | Random | - | 0.16 | 0.79 | 0.26 | 1.31 | 0.19 | 0.95 | 0.06 | 0.32 |
| Zero Shot | Pic2Word [47] | CC3M [9] | 26.2 | 43.6 | 20.0 | 40.2 | 27.9 | 47.4 | 24.7 | 43.7 |
| | CoVR-BLIP | - | 16.68 | 30.67 | 13.44 | 31.93 | 17.85 | 35.70 | 15.99 | 32.77 |
| | CoVR-BLIP | WebVid-CoVR | **30.37** | **46.27** | **21.81** | **39.02** | **30.85** | **49.06** | **27.68** | **44.78** |

Table 5: **Data size:** We experimentally validate the importance of the number of videos used for data generation and of filtering the generated data, evaluated by downstream performance on WebVid-CoVR$_m$ (test), CIRR (test), and FashionIQ (val). All models are trained for the same number of iterations on the generated data. Training batches are made up with distinct target videos.

| Initial #videos | Generated #target videos | Generated #triplets | Filtering | WebVid-CoVR$_m$ R@1 | WebVid-CoVR$_m$ MeanR | CIRR R@1 | CIRR MeanR | FashionIQ R@10 | FashionIQ MeanR |
|---|---|---|---|---|---|---|---|---|---|
| 0 | - | - | - | 15.98 | 37.44 | 19.76 | 45.88 | 15.99 | 24.38 |
| 200k | 10k | 4k | ✓ | 25.13 | 51.22 | 33.90 | 63.32 | 26.22 | 35.83 |
| 500k | 14k | 66k | ✓ | 46.04 | 74.24 | 38.31 | 67.80 | **28.76** | **37.78** |
| 1M | 38k | 269k | ✓ | 48.46 | 76.47 | 38.51 | 67.95 | 28.41 | 37.38 |
| 2.5M | 130k | 1.6M | ✓ | **54.87** | **80.57** | **38.55** | **68.55** | 27.68 | 36.23 |
| 2.5M | 212k | 3.6M | ✗ | 49.86 | 76.12 | 34.10 | 64.77 | 25.81 | 34.16 |

We also evaluate baselines that are not trained on WebVid-CoVR and that directly apply the pretrained BLIP model [31] to the CoVR task. These baselines outperform the random baseline but underperform compared to models trained on WebVid-CoVR, showing the benefit of our automatically generated training dataset. Note that BLIP [31] is pretrained for image-text retrieval but not for image-text-image retrieval, hence the drop in performance when applied directly to CoVR with both input modalities compared to only using visual information.

## 4.3 Transfer learning to composed image retrieval

While our focus is video retrieval, we also experiment with transferring our CoVR models to image retrieval tasks on standard CoIR benchmarks. We define zero-shot CoIR as not using any manually annotated CoIR triplet for training. We perform zero-shot CoIR by directly applying our model trained on our automatically generated WebVid-CoVR dataset to CoIR tasks and also explore finetuning our model on the training set of the downstream benchmark.

Tables 3 and 4 report results on CIRR and Fashion-IQ datasets, respectively. These results show that our model highly benefits from training on WebVid-CoVR, especially in the zero-shot setting, on both datasets. In addition, our model achieves state-of-the-art zero-shot performance on both CIRR and FashionIQ, and performs competitively in the finetuning setting on both benchmarks.

Table 6: **Modification text generation:** We compare our MTG-LLM to a rule-based MTG baseline and observe important gains in the downstream performance of the model trained on the generated data. All models are trained for the same number of iterations on the generated data.

| Model | WebVid-CoVR | | | | CIRR | | | |
|---|---|---|---|---|---|---|---|---|
| | R@1 | R@5 | R@10 | R@50 | R@1 | R@5 | R@10 | R@50 |
| Rule-based | 43.00 | 70.10 | 79.38 | 94.58 | 15.90 | 39.06 | 52.36 | 79.22 |
| MTG-LLM | **54.87** | **80.99** | **88.30** | **98.11** | **38.55** | **66.80** | **77.25** | **91.61** |

Table 7: **Ablations on training strategies:** Constructing batches of distinct target videos (and not CoVR triplets) and our hard negative mining both benefit the downstream CoVR/CoIR performance.

| Iteration | Hard negatives | WebVid-CoVR$_m$ | | | | CIRR | | | |
|---|---|---|---|---|---|---|---|---|---|
| | | R@1 | R@5 | R@10 | R@50 | R@1 | R@5 | R@10 | R@50 |
| Triplets | ✓ | 47.68 | 76.14 | 85.46 | 97.25 | 38.53 | 65.66 | 76.22 | 90.34 |
| Videos | ✗ | 54.00 | 80.53 | 88.01 | 98.03 | 38.34 | 66.75 | 77.21 | 91.42 |
| Videos | ✓ | **54.87** | **80.99** | **88.30** | **98.11** | **38.55** | **66.80** | **77.25** | **91.61** |

## 4.4 Ablation studies

In this Section, we ablate the importance of several key aspects of our method by evaluating the downstream performance of the model trained only on WebVid-CoVR.

**Importance of data scale.** In Table 5, we evaluate the importance of the scale of the dataset of video-captions used in our generation pipeline. We construct subsets of videos such that larger ones include smaller ones, and only keep triplets that contain the sampled videos for training. We find that results steadily increase when using more videos, demonstrating that our method largely benefits from scaling the size of the seed dataset of video-captions. We also observe the importance of the filtering techniques described in Section 3.1, as the model trained on unfiltered generated data underperforms.

**Modification text generation.** We use a large language model finetuned for modification text generation as explained in Section 3.1. We here compare this solution to a rule-based baseline that uses several templates to generate the modification text given the two captions that differ by one word. Specifically, the modification text is based on the two different words from the captions. We generate templates that use these words and chose one at random during training. These templates include variations such as *"Remove* txt_diff$_1$*"* and *"Change* txt_diff$_1$ *for* txt_diff$_2$*"*. A full list of all the templates can be seen in the Supplementary Material. In Table 6, we show that our large language model generates better modification texts than the rule-based baseline, by evaluating the results of the model trained on the generated data. Qualitative examples comparing the two approaches are provided in the Supplementary Material.

**Training strategies.** In Table 7, we first show the benefit on WebVid-CoVR of training by iterating on target videos instead of CoVR triplets. This is to avoid having the same target video appearing multiple times in a training batch, hence increasing the number of correct negatives that are used in the contrastive loss. Furthermore, sampling hard negatives, as described in Section 3.3, also slightly benefits the downstream performance.

## 5 Conclusions, Limitations, and Societal Impacts

In this work, we studied the new task of CoVR by proposing a simple yet effective methodology to create automatic training data. Our results on several benchmarks (including our manually curated video benchmark, as well as existing image benchmarks) suggest that, while noisy, such an automated and scalable approach can provide effective CoVR model training. One potential limitation of our method is that our dataset may not depict some visible changes due to the way we generate triplets. Moroever, our modification text generation model is suboptimal due to only inputting text (i.e., without looking at images). Future work can incorporate visually grounded modification generation.

**Societal impact.** Our model constitutes a generic multi-modal search tool, but is not intended for a specific application. While there are helpful use cases such as online shopping, traveling, and personal development (i.e., how-to), there may be potential privacy risks associated to surveillance applications, searching for a specific person in videos.

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
