# OpenReview forum: "CoVR: Learning Composed Video Retrieval from Web Video Captions"
_NeurIPS.cc/2023/Conference — Submitted to NeurIPS 2023_

### Official Review · Reviewer_xJ3a · 2023-07-04

**Soundness:** 3 good
**Presentation:** 3 good
**Contribution:** 4 excellent
**Rating:** 5
**Confidence:** 5

**Summary:**

This paper focuses on the task of Compositional Video Retrieval (CoVR) in which given a video and a text which modifies the video aims to rank and retrieve the modified video. In this work, a new large-scale dataset for pre-training (named WebVid-CoVR) is developed with generated modifications from a Large Language Model (LLM). Additionally, a smaller dataset for evaluation is manually annotated. Alongside this, a method is proposed that follows HH-NCE to learn the video compositions. Experiments show that the proposed method works well in both the supervised and zero-shot cases, and even translates well to Compositional Image Retrieval (CoIR) datasets CIRR and FashionIQ.

**Strengths:**

The created dataset(s) looks like it will be a useful addition to vision-language pre-training and especially for the Composed Video Retrieval task which currently doesn't have a dataset to train on.

The methodology of creating the dataset is well-done and all the steps make sense, especially for the manually annotated portion of the test set.

The paper is generally well-written and easy to follow.

**Weaknesses:**

Line 137: Is the Top K sampling referring to the tokens within a modification text? If not it's not clear what this means as only a single modification is generated as specified on Line 138.

Regarding the rule-based ablation in Table 6, could the generated rules be paraphrased by a LLM? In this case, the MTG-LLM wouldn't necessarily need to be fine-tuned and a standard LLM could be used instead.

Line 164: The paper mentions that WebVid-CoVR is noisy which is why a subset is manually annotated for the test set. What is the main source of noise within the main dataset? Is this from the generated modifications? The captions not matching the videos or something else? Has a human study shown how much of the dataset can be considered noisy from looking at a small number of samples?

\alpha and \tau are learnable parameters? Normally, \tau at least, is a set hyperparameter. How these are learnt aren't mentioned within the text.

Has any analysis been performed into the types of modifications within the dataset? I.e. noun/adjective/verb changes and others? It is mentioned within the limitations that certain modifications may not have been generated and it would be good to see what coverage there could be - even if this is from a human annotated subset.



**Questions:**

For more context regarding the questions, please see the weaknesses section above.
1. Has any analysis been performed into the types of modifications within the dataset?
2. What is the main source of noise within the main dataset?
3. How are \alpha and \tau learnt?
4. How does learning \tau compare to keeping it a fixed value as is normally done in practice? How does \tau change during training?
5. Could the generated rules be paraphrased by a LLM? Would this change the performance within Table 6?
6. Is the Top K sampling referring to the tokens within a modification text?

**Limitations:**

There is a short section on limitations which mentions the scope of modifications within the dataset, I think this could be improved by performing some analysis on the generated prompts.

---

> ### Author Rebuttal · Authors · 2023-08-09
>
> **1. Has any analysis been performed into the types of modifications within the dataset?**
> We analyze the number of words in the modification text in Figure A.2 of the supplementary material, but in Figure R.3 of the rebuttal PDF we provide further analysis on the distribution of noun/adjective/verb as suggested by the reviewer. We also include a visualization of the verb-noun frequency heatmap in Figure R.4, which provides insights into the distribution of verb-noun count combinations across our dataset.
>
> We also conducted an analysis using part-of-speech tagging on the captions. The resulting visual representation, presented in Figure R.6 of the rebuttal PDF, illustrates the transition of POS tags across the difference words in Caption 1 and Caption 2.
>
> Finally, indeed, we mentioned within the limitations that certain modifications may not have been generated. We give examples for a couple such scenarios in our response to `Reviewer #4ap2` with the title *“4. Examples of other types of modifications not captured by single-word differences”*.
>
> ---
> ---
>
>
> **2. What is the main source of noise within the main dataset?**
> As mentioned in L178, about 23% of the automatic collection can be considered as noisy, because this was the percentage of discarded triplets when manually curating the WebVid-CoVR$_m$ test set. We expect a similar noise ratio in the training set.
>
> To address the reviewer’s question more in detail, we manually went over the triplet examples that were marked as unsuitable (therefore discarded) when annotating test set. We marked whether the reason for discarding falls within any of the following categories, and computed the following percentages (normalized by the number of discarded triplets).
>
> * 35%: The generated modification text does not describe the visual difference. Primarily attributed to either the quality of the video captions or the output generated by the MTG-LLM.
> * 28%: Paired videos are visually too similar.
> * 15%: Paired videos are visually too different.
> * 13%: At least one of the videos is difficult to understand/low quality.
> * 9%: Captions are too similar (e.g., one-word difference doesn’t change the meaning: “On the chairlift” and “Ride the chairlift”).
>
> While the first category of errors is the largest, it is important to also note that our strict standards for the test set necessitated the discarding of many triplets that could potentially be useful for training.
>
> ---
> ---
>
> **3. How are $\alpha$ and $\tau$ learnt?**
> We use $\tau$ in the same way as BLIP, i.e., as a learnable parameter clamped between 0.001 and 0.5 with initial value of 0.07. For $\alpha$, we set the initialize the learnable parameter with 0.1. We will mention this in the text.
>
> ---
> ---
>
> **4. How does learning $\tau$ compare to keeping it a fixed value as is normally done in practice? How does $\tau$ change during training?**
> It is worth noting that the adoption of learned temperature parameters is not unprecedented. Both the BLIP and CLIP models also incorporate a learned temperature parameter as part of their architectures. The initial value of $\tau$ is set to 0.07 and it decreases monotonically down to **0.06794**, i.e., in practice the value does not change much. We suspect that the scale of the parameter updates is not large enough. To mitigate the small updates, we also investigate learning $\tau / 100$ and then multiplying the learned value by 100 to have more granularity. If we do this, we initialize $\tau$ at 0.0007*100= 0.07 and it decreases until 0.0003835 * 100 = **0.03835** over training iterations and it stabilizes. We plot the curve in Figure R.5 of the rebuttal PDF, and report the WebVid-CovR results with/without learning $\tau$, as well as learning with this division trick.
>
> | Tau | R@1 | R@5 | R@10 | R@50 |
> |---|---|---|---|---|
> | Not learned | 54.70 | 81.07 | 88.42 | 98.11 |
> | Learned | 54.87 | 80.99 | 88.30 | 98.11 |
> | Learned (/100) | **55.15** | **81.27** | **89.28** | **98.28** |
> Table R.4
> ---
> ---
>
> **5. Could we paraphrase the rule-based generated texts with a standard LLM to avoid finetuning MTG-LLM?**
> We thank the reviewer for the suggestion. To answer this question, we investigated the possibility of paraphrasing the generated rule-based modification texts using an LLM. Initially, we experimented with LLaMA and LLaMA 2 for paraphrasing, but the results were qualitatively unsatisfactory. For instance, the output generated by LLaMA was overly verbose and not suitable for the CoVR task. However, we found that employing gpt-3.5-turbo from OpenAI yielded significantly improved paraphrased responses.
>
> Paraphrasing the rule-base examples significantly boosts the results (from 43 to 53 R@1) at the cost of running an expensive LLM ($43 cost for this experiment for 1 paraphrasing per modification text on the entire dataset). On the other hand, our finetuning of the MTG-LLM, which is highly cost-effective (only 1 epoch and 715 text examples), leads to overall better results.
> | Type | R@1 | R@5 | R@10 | R@50 |
> |---|---|---|---|---|
> | Rule based | 43.00 | 70.10 | 79.38 | 94.58 |
> | Rule based paraphrased with GPT-3.5 | 53.45 | 79.64 | 87.19 | 97.70 |
> | Finetuned MTG-LLM with LLaMA | **54.87** | **80.99** | **88.30** | **98.11** |
> Table R.5
>
> Given these observations, we believe that finetuning the MTG-LLM is a preferable approach, as it outperforms gpt-3.5-turbo in terms of cost-effectiveness.
>
> ---
> ---
>
> **6. Is the Top K sampling referring to the tokens within a modification text?**
> Yes, top-k sampling refers to the tokens within a modification text. As described and introduced in Fan et al. [2018], top-k sampling can generate more diverse outputs than beam search thanks to its randomness (e.g., “beam search produces common phrases and repetitive text from the training set”). However, the reviewer is correct that we only generate one modification text per caption pair in our study. We will clarify L136 by removing *“To increase the diversity of the generated samples”*.

---

> > ### Comment · Reviewer_xJ3a · 2023-08-14
> > **Thank you for the clarifications**
> >
> > Thank you for providing detailed responses to my initial review and addressing my questions. After reading these and the other reviewers' comments I am still in favour of accepting this paper.

---

### Official Review · Reviewer_4ap2 · 2023-07-05

**Soundness:** 3 good
**Presentation:** 3 good
**Contribution:** 2 fair
**Rating:** 5
**Confidence:** 4

**Summary:**

The paper addresses the challenge of composed video retrieval, which involves querying a video and a modification text to find videos that exhibit similar visual characteristics with the desired modification. The main challenge is the lack of data for composed video retrieval. To overcome this, the paper proposes mining paired videos with similar captions from a large database and generating the corresponding modification text using a large language model. The paper explains the BLIP-based video and text encoder and the training process for the model using the collected data. Experimental results demonstrate that the model trained on the compiled dataset can generalize to both zero-shot and fine-tuning settings.

**Strengths:**

- The paper introduces a novel task of video retrieval and establishes a benchmark for future research in this area.
- Overall, the paper is well-written, clear, and easy to follow.

**Weaknesses:**

- Examples in Figure 3 suggest that most samples do not require handling dynamic content, implying that there may not be a significant difference between the proposed CoVR task and existing CoIR tasks.
- The MTG-LLM method requires manually created data for fine-tuning, which can be resource-intensive.
- The training method is standard and not particularly innovative, although it is reasonable for the task.
- The training data is limited to modifications that can be represented with single-word differences, potentially excluding other types of modifications. This point is mentioned as a limitation in the paper. By providing concrete examples of scenarios not addressed by this work, readers will understand the challenge clearly.

**Questions:**

- What annotations are added to the dataset? (line 145)

**Limitations:**

The paper mentions two limitations.
- The data creation pipeline may not adequately capture some visible changes due to its design.
- The generation of modification text may not be optimal as the text generation depends only on input captions.

I appreciate the authors for pointing out the meaningful limitations and suggesting ideas for future research.

---

> ### Author Rebuttal · Authors · 2023-08-09
>
> **1. Dynamic vs static content in CoVR.**
> We acknowledge that, as the reviewer pointed out, some videos can be retrieved by only looking at one frame. This is a common problem in video datasets as highlighted by Jie et al. [“Revealing Single Frame Bias for Video-and-Language Learning”, ACL 2023]. However, our results in Table 2 show that using multiple frames benefits the CoVR performance.
>
> Here, we conduct a further analysis on the training set of WebVid-CoVR based on optical flow to detect the frequency of static videos in our data. We computed the optical flow using the Gunnar Farneback's algorithm and empirically chose a magnitude threshold of 1 to distinguish between videos with static and dynamic elements. The magnitude value is obtained by averaging the Euclidean norms of motion vectors in both horizontal and vertical directions across the computed video frames. We identified that around 25% of the triplets contain static target videos, which represents approximately 21% of the overall target videos. In the table below, we show a minor decrease in performance if we omit these static videos during training (while maintaining the same iteration count). This may be because image training data can still be complementary to video training [4].
>
> In Figure R.2 of the rebuttal PDF, we also illustrate visual examples to motivate the use of multiple frames. We show two videos (one detected to be static based on optical flow, the other dynamic), where multiple frames are necessary to associate the visual content to the video caption (e.g., for the “timelapse” concept, and the action “hiding”).
>
> |                | Percentage of data | R@1   | R@5   | R@10  | R@50  |
> |----------------|--------------------|-------|-------|-------|-------|
> | Static         | 25                | 50.99 | 77.46 | 85.92 | 97.45 |
> | Dynamic        | 75                | 54.11 | 80.46 | 86.95 | 97.82 |
> | Static+Dynamic | 100               | **54.87** | **80.99** | **88.30** | **98.11** |
> Table R.3: Training with static or dynamic partitions of WebVid-CoVR.
>
> ---
> ---
>
> **2. MTG-LLM requires manually created data for finetuning.**
> While it is true that our approach involves using 715 text triplets, it's important to note that we repurpose pre-existing data created by InstructPix2Pix in the context of text-conditioned image editing [8]. Furthermore, the data creation process for text triplets is quite fast considering it involves only 715 text samples.
>
> It is worth noting that, despite the manual data we used for the MTG-LLM, we provide two alternatives where no manually data is required: a rule-based approach in Table 6 and a prompting technique without finetuning in Table A.3. However, the rule-based and prompting methods exhibit lower results in comparison to MTG-LLM, highlighting the effectiveness of finetuning the MTG-LLM. We anticipate that future advancements in language models may enhance the effectiveness of the prompting technique further, removing the need to finetune.
>
> ---
> ---
>
> **3. Training method is standard.**
> We agree with the reviewer that the training method is standard, we do not claim novelty in this component, and on the contrary build our focus towards the data generation methodology. To reiterate, our contribution lies in the development of a scalable approach that enables the automatic generation of CoVR training data directly from Web video captions. This innovative approach not only facilitates efficient data collection but also addresses the challenge of acquiring diverse and relevant training samples. Additionally, we have introduced a new manually-curated benchmark to evaluate the CoVR task.
>
> ---
> ---
>
>
> **4. Examples of other types of modifications not captured by single-word differences.**
> As requested by the reviewer, we provide concrete examples of scenarios potentially excluded due to detecting one-word difference between caption pairs.
>
> * Multiple modifications: We cannot find examples where multiple aspects change at once. For example, the following example from the CIRR dataset is not captured, as two things are changed between query and target image: “The target photo is a close up of a similar dog, but it is swimming on its own with a tennis ball in its mouth.”
> * In the following example, the difference between the two captions is more than one word (“empty”, “and kids playing”), so we discard this pair. However the target could be formulated as “add kids playing”.
>   * $caption_1$: An empty park with green trees.
>   * $caption_2$: A park with green trees and kids playing.
>
> We would like to mention that, in preliminary analysis, we also explored pairing captions via text embedding similarity (instead of one-word difference), but our qualitative results showed that such similarity metric is too noisy to be used to detect caption pairs that differ by an easily describable modification. For example, some would require also checking the visuals to determine the modification (e.g., whether the park is empty in both videos in the above example). Instead, with our simple approach, we avoid this noise, and already obtain more than a million triplets.
>
> ---
> ---
>
> **5. What annotations are added to the dataset?**
> The full list of added examples can be seen in Table A.2 of the supplementary material. We will add a pointer from L145.

---

> > ### Comment · Reviewer_4ap2 · 2023-08-13
> > **Thanks for your responses**
> >
> > Thank you for your responses.
> > The rebuttal addressed my concerns, and I remain positive about this paper.

---

### Official Review · Reviewer_xqD8 · 2023-07-11

**Soundness:** 3 good
**Presentation:** 3 good
**Contribution:** 3 good
**Rating:** 5
**Confidence:** 3

**Summary:**

This paper proposes a scalable approach to automatically generate composed visual retrieval training data. Specifically, based on the WebVid2M dataset, the authors generates a WebVid-CoVR training dataset with 1.6M CoVR triplets.

**Strengths:**

The data augmentation strategy is scalable.

**Weaknesses:**

The overhead for the dataset augmentation should be detailed.

**Questions:**

The overhead for the dataset augmentation should be detailed.

**Limitations:**

The overhead for the dataset augmentation should be detailed.

---

> ### Author Rebuttal · Authors · 2023-08-09
>
> **What is the dataset augmentation overhead?**
> We outline the detailed computation time for each step of the dataset generation. The computation times below are obtained using a *single* NVIDIA RTX A6000, but it is important to note that most of the processes can be parallelized, which would significantly reduce the wallclock time required. In practice, we used 2 GPUs.
> * **1. Text embedding extraction:** We extracted text embeddings from 2 million distinct captions out of a total of 2.4 million video-caption pairs. This process completed in less than **2 hours**.
> * **2. Caption similarity search:** To identify captions with one-word differences, we employed the faiss library [Johnson et al. 2019] to select the 100 closest captions, avoiding the need to compare each caption against the entire set of 2 million captions. This optimization significantly reduced the search time, resulting in **2.5 hours**.
> * **3. Text similarity filtering:** Thanks to the precomputed text embeddings, the text similarity filtering step incurred no additional time overhead. All the text filtering processes were completed in **less than 5 minutes**, even on a large pool of 1.2 million captions.
> * **4. Video similarity computation:** To filter by video similarity, we extracted the middle frame from approximately 135,000 videos and computed CLIP embeddings. This step takes approximately **3 hours**.
> * **5. MTG-LLM model finetuning:** Finetuning for 715 examples takes **less than 10 minutes**. Note that the time required to finetune the MTG-LLM model is independent of the number of CoVR triplets we generate.
> * **6. Modification text generation:** This is the most time-consuming stage of the pipeline. To optimize its speed, we modified the original ``Lightning-AI/lit-llama`` implementation on github to enable batch inference. This step takes around **24 hours** to process the 1.6 million caption pairs.
>
> This analysis demonstrates the feasibility and efficiency of our approach for the current dataset. We will add this breakdown to our supplementary material.

---

### Official Review · Reviewer_k4Sb · 2023-07-16

**Soundness:** 3 good
**Presentation:** 3 good
**Contribution:** 3 good
**Rating:** 5
**Confidence:** 4

**Summary:**

This paper automatically constructs a new dataset called WebVid-CoVR by applying a scalable automatic dataset creation procedure that generates triplets from video-caption pairs to a large-scale WebVid2M collection, resulting in 1.6M triplets. Moreover, this paper introduces a new benchmark for composed video retrieval(CoVR) and contribute a manually annotated evaluation set, along with baseline results. The results demonstrate that training a CoVR model on WebVid-CoVR transfers well to CoIR with competitive performance.

**Strengths:**

- The automatic triplet generation pipeline is carefully designed with many phases.
- Provides strong baseline results on CoVR and shows transferability to CoIR.

**Weaknesses:**

- The automatic triplet generation pipeline seems to rely only on caption similarity, as well as MTG-LLM, which may introduce noise and ignore visual similarity. Visual similarity between videos should also be taken into account.
- More dataset analysis, especially about the visual part, should be provided to gain more understanding of the characteristics of WebVid-CoVR. Besides, the human check may be necessary.
- Only one model (CoVR-BLIP) is evaluated on the proposed CoVR task. More baselines like frozen/finetuned multimodal transformers could be added.

**Questions:**

Please refer to Weaknesses.

**Limitations:**

Please refer to Weaknesses.

---

> ### Author Rebuttal · Authors · 2023-08-09
>
> **1. Incorporating visual similarity between videos.**
> We agree that considering visual similarity between videos is important for the triplet generation process. However, in this work, we indeed rely only on caption similarity because, as mentioned in L114, we rely on the assumption that caption similarity implies visual similarity. Another reason we chose not to rely on visual similarity was to avoid creating an undesired bias in the dataset. For example, if we restricted *all* the video pairs in triplets to have similarity within a range, this would potentially lead the model to learn to ignore the input modification text, and only focus on the visual query.
>
> On the other hand, we note that, we already incorporate visual similarity to some extent in some cases. As mentioned in L152, when multiple paired videos match the same video caption, to avoid over-representing certain modification texts, we select the top 10 pairs with the highest visual similarity. By doing so, we effectively reduce the number of possible triplets from 2.4M to 1.6M, discarding 800k triplets.
>
> Moreover, we performed an additional experiment. We train on WebVid-CoVR triplets whose video pairs fall above a certain visual similarity threshold. The table below summarizes the results of evaluating on the zero-shot CoIR benchmarks. We observe that the performance consistently decreases with increased threshold, while also filtering out a large portion of the training data.
>
> | Visual Sim. Threshold | Percentage of data | CIRR R@1 | FashionIQ R@10 mean |
> |---|---|---|---|
> | 0.00 (None) | 100% | **38.55** | **27.68** |
> | 0.55 | 92% | 38.07 | 26.81 |
> | 0.60 | 83% | 37.69 | 25.75 |
> | 0.65 | 71% | 36.96 | 25.77 |
> | 0.70 | 55% | 35.49 | 23.72 |
> Table R.1: We observe worse performance on CoIR zero-shot benchmarks as we increase the visual similarity threshold in our training data. We train each model for the same number of iterations.
>
> ---
> ---
>
> **2. More dataset analysis.**
> We provide dataset statistics in Section A of the supplementary material with (i) the text/video similarity of the caption/video pairs in Figure A.1, (ii) the histogram of the number of words in the generated modification text in Figure A.2, and (iii) the distribution of number of triplets per target video in Figure A.3.
>
> As per reviewer’s request, we further provide more analysis. We plot the distribution of video categories in Figure R.1 of the rebuttal PDF. These categories are found using the WebVid metadata provided by the recent *shinonomelab/cleanvid-15m_map* on HuggingFace Datasets. We find 50% of WebVid-CoVR videos in this metadata collection. Note more than one category can be associated with a single video (e.g., Nature and Animals/Wildlife for a video of a fish in the ocean). We further point to our response to `Reviewer #xJ3a` for more analysis on the types of modifications and our human-checked test set.
>
> ---
> ---
>
> **3. More baselines for the CoVR task.**
> As suggested by the reviewer, here, we include additional baselines:
> * LF-CLIP (MLP): We implement the architecture of Combiner [7], referred to as late fusion by CASE [28]. In detail, we concatenate the CLIP visual and text features (from the visual query and the modification text). The combined multimodal representation is then learned on WebVid-CoVR with an MLP initialized randomly.
> * LF-CLIP (avg): We also implement a simpler late fusion as a baseline for the case we *do not* train on WebVid-CoVR. We average the CLIP visual and text features as our combined query representation.
> * LF-BLIP (MLP): Similar to [28], we implement the BLIP equivalent of the above LF-CLIP (MLP) baseline, using BLIP visual and text features instead of the CLIP ones. Note that we throw away the cross-attention layers and instead use an MLP to combine modalities.
> * LF-BLIP (avg): Here, we simply average the BLIP visual and text features as above.
> * Variants of pretrained BLIP models: We experiment with various pretrained BLIP models from [31], including the base BLIP, BLIP finetuned on Flickr30k, and BLIP finetuned on COCO (what we used in the paper). Note that, in this case, we use the existing cross-attention layers of BLIP as our multimodal combined representation and finetune them with WebVid-CoVR.
>
> The table below summarizes the results. We observe that the inclusion of LF-CLIP (avg) and LF-BLIP (avg) baselines notably enhance the initial zero-shot performance in the paper. Additionally, we observe that the BLIP model finetuned on COCO has the highest performance.
>
> | Model | Train on WebVid-CoVR | R@1 | R@5 | R@10 | R@50 |
> |---|---|---|---|---|---|
> | LF-CLIP (avg) | No | 19.91 | 39.57 | 47.33 | 67.49 |
> | LF-BLIP (avg) | No | 46.84 | 72.09 | 80.79 | 93.97 |
> | LF-CLIP (MLP) | Yes | 41.26 | 69.83 | 79.93 | 94.95 |
> | LF-BLIP (MLP) | Yes | 52.01 | 76.40 | 84.32 | 96.14 |
> | BLIP base | Yes | 51.89 | 79.76 | 87.11 | 97.70 |
> | BLIP ft Flickr30k | Yes | 54.11 | 80.46 | 87.44 | 97.99 |
> | BLIP ft COCO | Yes | **54.87** | **80.99** | **88.30** | **98.11** |
> Table R.2: Additional baselines on WebVid-CoVR.

---

### Author Rebuttal · Authors · 2023-08-09

We thank all four reviewers (`#k4Sb`, `#xqD8`, `#4ap2`, `#xJ3a`) for constructive feedback.
It is encouraging to see that our automatic triplet generation pipeline has been well-received, particularly for its careful design and multiple phases (`#k4Sb`, `#xJ3a`), as well as its scalability (`#xqD8`). Additionally, we are pleased that our Composed Video Retrieval (CoVR) task and dataset were recognized as having potential for future research (`#k4Sb`, `#xJ3a`), and that our experiments have showcased the adaptability of our approach to the CoIR task (`#k4Sb`), as indicated by the ability to generalize in both zero-shot and finetuning scenarios (`#4ap2`). We also appreciate the kind feedback on the readability and clarity of our paper (`#4ap2`, `#xJ3a`).
We address each of their comments individually and will update the paper accordingly.

---

### Comment · Area_Chair_c9ZF · 2023-08-17
**Please respond to authors' rebuttal**

Dear Reviewers,

Thanks for your contributions! Please respond to the authors after carefully reading their rebuttals and other reviews. If your assessment of the paper changes, please update your score with a short justification for the new rating.

The paper received diverging initial reviews. Please consider discussing with the authors or other reviewers to see whether we can reach a consensus.

Thank you,

AC

---

### Decision · Program_Chairs · 2023-09-21

**Decision:**

Reject

**Comment:**

This paper initially received mixed reviews. The main concerns are: 1) lack of comprehensive dataset evaluation results; 2) overhead for the dataset augmentation; 3) the limitations of the model design; and 4) some unclear presentation. In the rebuttal, although the authors have provided very detailed responses to these questions, there is still no reviewer champion for this submission. Thus, I recommend rejecting this paper. Meanwhile, I hope the authors can contain all these discussions for future resubmission, which will make it much stronger.